# LAIA-SQL: Enhancing Natural Language to SQL Generation in Multi-Table QA via Task Decomposition and Keyword Extraction

## Abstract

Natural Language to SQL (NL2SQL) provides an effective solution for multi-table question answering (Table QA) to automate data retrieval by transforming simple user queries into SQL commands. It enhances data accessibility and decision-making processes across various industries. Large Language Model (LLM) based NL2SQL methods have been shown to outperform rule-based or neural network-based NL2SQL methods. However, existing LLM-based NL2SQL approaches face challenges like inaccurate interpretation of user questions, slow retrieval speeds, erroneous SQL generation, and high operational costs. As there is a lack of datasets specifically designed to evaluate natural language understanding (NLU) in NL2SQL tasks and no models optimized for user question understanding in Table QA, we introduce **LAIA-NLU**, a novel dataset that dissects NLU into task decomposition and keyword extraction. LAIA-NLU contains 1,500 high-quality QA pairs, created through manual review. Using this dataset, we developed **LAIA-NLUer**, which is capable of effectively interpreting user intent in table-based queries. To further enhance NL2SQL performance in terms of speed, cost, and accuracy, we also present **LAIA-SQL**, a retrieval-augmented based NL2SQL framework. Experimental results show that LAIA-SQL outperforms state-of-the-art models, achieving an accuracy improvement to 67.28% in BIRD dataset, a 52.4% reduction in runtime, and a 97% decrease in operational costs. These improvements demonstrate the potential of our approach to advance multi-table data retrieval and analysis. Our code, dataset, and model will be publicly available to encourage further research in this field.

## 1 Introduction

Table Question Answering (Table QA) is a task to help users who are not proficient in coding skill or advanced spreadsheet software retrieve complex table data by question answering Javaid et al. (2023); Al Naqbi et al. (2024). A leading approach in Table QA is Natural Language to SQL (NL2SQL), which translates natural language queries into SQL, allowing users to interact with databases in everyday language Gao et al. (2023).

Recent research shows that NL2SQL methods leveraging Large Language Models (LLMs) outperform other rule-based or neural network based methods significantly Zhang et al. (2024a). A direct approach is prompting LLMs like GPT-4o OpenAI (2024c) to perform related tasks. However, this method often results in SQL statements with logical errors, inaccurate field recognition, and difficulty managing multi-table relationships Liu et al. (2024). We hypothesize that these issues arise from LLM's inadequate understanding of user question in Table QA scenarios.

Effective SQL generation requires the model to excel in natural language understanding (NLU), which can be divided into two areas: 1) fine-grained task decomposition and 2) precise keyword extraction. While the former is crucial for complex multi-table reasoning, the latter ensures accurate recognition of table and column names. However, as shown in Figure 1, using LLMs like GPT-4o for task decomposition and keyword extraction still presents challenges, as models may generate insufficient tasks and misidentify keywords. Addressing these challenges requires specialized training because it involves understanding and manipulating structured data within a specific context,

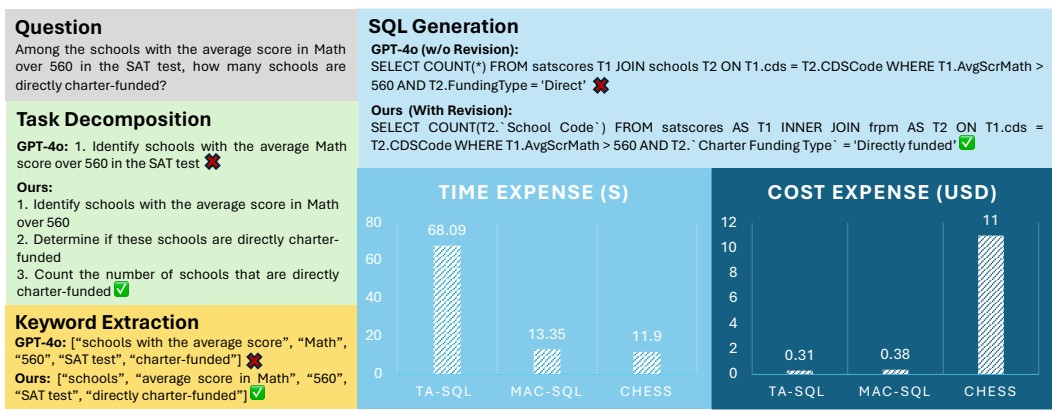

Figure 1: Comparison of advanced NL2SQL methods with LAIA-SQL. GPT-4o suffers from incomplete task decomposition and incorrect keyword extraction. Missing a revision module, GPT-4o shows lower code generation accuracy. Methods like MAC-SQL, CHESS, TA-SQL are efficient in either time or cost, but not both.

which is different from more general natural language tasks. Fine-grained task decomposition involves breaking down complex queries into smaller, precise steps aligned with the relational schema of databases. Precise keyword extraction requires accurately mapping natural language to specific table and column names, necessitating an intimate understanding of the database structure. Additionally, there is a lack of quantitative evaluation metrics for assessing NLU performance across different LLMs within the Table QA domain, which impedes progress in this specialized area.

Beyond directly applying large language models (LLMs) for NL2SQL, hybrid methods that combine LLMs with various modules have also shown promise. Notable examples include CHESS Talaei et al. (2024), TA-SQL Gao et al. (2023), and MAC-SQL Wang et al. (2023). Nevertheless, as demonstrated in Figure 1, challenges such as slow data retrieval, erroneous SQL code generation, and high operational costs still remain.

To systematically improve the field, we present three main contributions: (1) LAIA-NLU, a dataset specifically designed to evaluate natural language understanding (NLU) within NL2SQL methods, (2) the LAIA-NLUer model, optimized for Table QA, and (3) LAIA-SQL, a framework enhancing NL2SQL performance in accuracy, efficiency, and cost.

The LAIA-NLU dataset comprises 1,500 high-quality QA pairs focusing on task decomposition and keyword extraction. Derived from the BIRD dataset Li et al. (2024c), it has undergone three meticulous rounds of manual annotation. Leveraging LAIA-NLU, we introduce LAIA-NLUer, a model fine-tuned based on GPT-4o-Mini. We assessed the performance of LAIA-NLUer by comparing it to six foundational models, using BLEU Papineni et al. (2002), ROUGE Lin (2004), and GPT-4o scores for task decomposition and F1 scores for keyword extraction. Our observations indicate that models fine-tuned with larger base models like GPT-4o-Mini excel at task decomposition, while smaller base models like Mistral-7B outperform in keyword extraction. Furthermore, results show that LAIA-NLUer fine-tuned with GPT-4o-Mini significantly enhances NL2SQL capabilities, drastically improving SQL generation accuracy compared with all other base models.

Lastly, we propose LAIA-SQL, an agent framework refined from CHESS Talaei et al. (2024). Through ablation studies, LAIA-SQL has been optimized into three main modules: User Question Understanding (UQU), Entity Retrieval, and Generation. In this study, we used LAIA-NLUer for the UQU module to enhance comprehension, combined retrieval and re-ranking in the Entity Retrieval module for improved accuracy, and introduced a revision process guided by task reasoning and error feedback during code generation. Experimental results demonstrate that this instance of LAIA-SQL outperforms all state-of-the-art open-source NL2SQL methods, achieving 67.28% accuracy on the BIRD dev dataset and 88.7% accuracy on the Spider dev dataset. LAIA-SQL also boasts substantially faster processing, answering 10 questions in just 56.81 seconds at a cost of $0.32, with an 80% accuracy rate. Compared to the leading NL2SQL methods using GPT-4o, LAIA-SQL reduced

runtime by 52.4% and operational costs by 97%, while maintaining the highest accuracy among advanced open-source NL2SQL methods.

## 2 RELATED WORK

### 2.1 TABLE QUESTION ANSWERING

The field of Table QA aims to deliver accurate answers derived from table data through precise and effective reasoning techniques. Initial approaches emphasized discrete reasoning Jin et al. (2022), with notable efforts like TAT-QA Zhu et al. (2021), FinQA Chen et al. (2021), and MVGE Ma et al. (2017) employing internal context learning (ICL), fine-tuning, and pre-training methods. These made significant strides but struggled with adaptability in multi-table scenarios Zhang et al. (2024b). Recently, methods have evolved to convert tabular data into graph structures for enhanced reasoning, as seen with GraphRAG Edge et al. (2024). Despite their promise, these methods remain time-consuming, resource-intensive, and face challenges in accurate graph construction Yu et al. (2024).

In parallel, NL2SQL research, which translates natural language questions into SQL queries, offers a more efficient and cost-effective solution Gao et al. (2023). NL2SQL technologies are mainly categorized into rule-based, neural network-based, Pre-trained Language Models (PLM)-based, and Large Language Models (LLM)-based approaches Li et al. (2024a). Initially, rule-based approaches prevailed, utilizing predefined rules or semantic parsers Katsogiannis-Meimarakis & Koutrika (2021), but were soon superseded by more scalable neural network techniques. By 2017, PLM methods, particularly those employing models like BERT Devlin (2018), took precedence. Currently, LLMs, exemplified by GPT-4 Achiam et al. (2023), dominate the field, powering advanced methods such as CHESS Talaei et al. (2024), DAIL-SQL Gao et al. (2023), and MAC-SQL Wang et al. (2023). These advanced methods feature specialized modules like filters, evaluators, and self-correction mechanisms to refine their outputs. Despite their sophistication, LLM-based methods still grapple with challenges like low accuracy, high operational costs, and significant runtime, constraining their practical utility Li et al. (2024a).

### 2.2 NATURAL LANGUAGE UNDERSTANDING

Natural Language Understanding (NLU) is a cornerstone of AI, enabling machines to interpret and process human language Allen (1988). This field encompasses a wide range of tasks, from keyword extraction to complex question answering Yu et al. (2023). The advent of LLMs like Gemini-Pro Reid et al. (2024), GPT-4 Achiam et al. (2023), and Mistral Jiang et al. (2023) has revolutionized NLU, pushing the boundaries of machine comprehension.

To further enhance NLU capabilities, researchers have investigated various innovative methods. These include sophisticated text alignment Zha et al. (2024), the integration of human-written explanations Liu et al. (2021), and advanced reasoning techniques like Chain of Thought (COT) Wei et al. (2022), Tree of Thought Yao et al. (2024), and Buffer of Thought Yang et al. (2024b). Specialized datasets such as Adversarial NLI Nie et al. (2019) and SemEval-2024 Task 2 Jullien et al. (2024) have been created to evaluate and refine LLMs' NLU proficiency.

Despite these advancements, substantial challenges persist in NLU, especially in table QA. While large language models (LLMs) exhibit impressive reasoning capabilities, they often struggle with precise information extraction and reasoning from tabular data. A crucial limitation is their inability to distinguish between meaningful and nonsensical language in user queries, and to consistently identify and extract relevant keywords corresponding to filter values, column names, or table names in a database. This deficiency underscores the pressing need for specialized datasets and fine-tuned models tailored specifically for NLU in table QA.

## 3 LAIA-NLU DATASET CREATION

As illustrated in Figure 1, current LLMs demonstrate limited NLU capabilities in table QA, adversely affecting the final accuracy of NL2SQL. Furthermore, there are no existing datasets to evaluate these models in terms of NLU within table QA. To address this gap, we introduce the LAIA-NLU dataset.

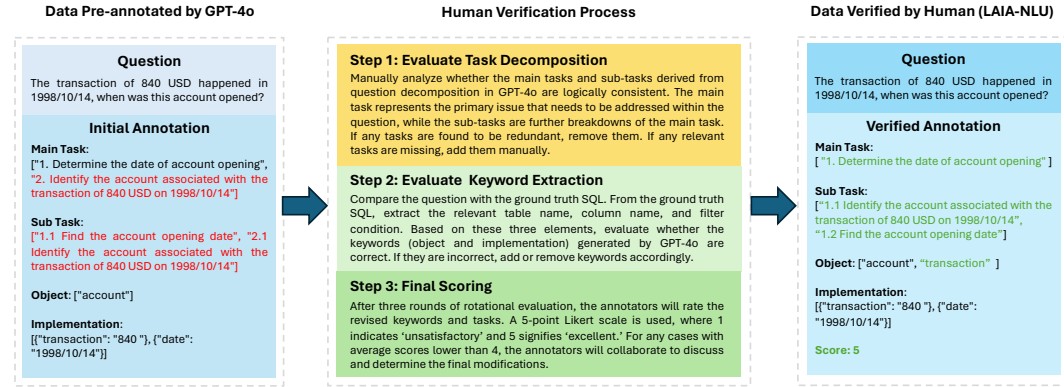

Figure 2: Dataset creation process of LAIA-NLU. GPT-4o firstly generates tasks, sub-tasks, objects, and implementations from user questions. Human annotators then verify and modify the task decomposition and keyword extraction for accuracy. After three rounds of cross-validation and final scoring, low-scoring results are reviewed and refined by discussion, producing LAIA-NLU.

## 3.1 DATA SOURCES

LAIA-NLU was derived from the BIRD dataset Li et al. (2024c) for its validated origins and extensive research use. BIRD comprises 12,751 text-to-SQL pairs across 95 databases, totaling 33.4 GB and spanning 37 professional domains, designed specifically for evaluating and training NL2SQL methods. It integrates 80 open-source relational databases from platforms like Kaggle and Relation.vit. To prevent data leakage, 15 additional relational databases were created for a hidden test set. The BIRD team used crowdsourcing to collect natural language questions paired with corresponding SQLs. Given its broad, validated origins and extensive research use, BIRD was chosen as our data source.

## 3.2 SELECTION AND ANNOTATION

We randomly selected 1,500 instances from the BIRD dataset's training data Li et al. (2024c). Each instance comprises a user question and the corresponding ground truth SQL query. Initially, we employed GPT-4o OpenAI (2024c) to perform task decomposition and keyword extraction. As illustrated in Figure 2, task decomposition involved breaking down the user question into two components: the main task and sub-tasks. The main task represents the primary goal derived from the user question, while sub-tasks further refine the main task. In the keyword extraction phase, keywords were categorized into two types: object and implementation. The object category includes terms related to table and column names in the user question, while implementation involves filtering criteria represented by a dictionary, where the keys denote filtering actions and the values specify the conditions. These elements collectively facilitate similarity matching within the database.

However, despite implementing Chain of Thought (CoT) Wei et al. (2022) and few-shot techniques Brown (2020), GPT-4o's performance in interpreting user queries was suboptimal. As shown on the left side of Figure 2, GPT-4o often produced redundant or incomplete tasks and extracted incorrect keywords. This necessitated manual refinement of the generated raw data.

Therefore, we invited three expert annotators to review and correct GPT-4o generated data. Our annotation strategy entailed a three-phase cyclic process to ensure cross-validation and accuracy. Each annotator began with different subsets (A, B, C) before Phase 1, exchanging and reviewing modified subsets in subsequential phases until all data was thoroughly evaluated by all annotators. As depicted in the Human Validation Process in Figure 2, the three-step process follows:

**1. Evaluate Task Decomposition** Annotators first reviewed each question manually to assess the accuracy of the main tasks and sub-tasks generated by GPT-4o. They checked for logical consistency, removed redundant tasks, and added any missing relevant tasks manually.

**2. Evaluate Keyword Extraction** Keywords were categorized into objects and implementations. Annotators compared the keywords generated by GPT-4o with the user questions and corresponding ground truth SQL elements (such as filters, table names, and column names), to ensure accuracy.

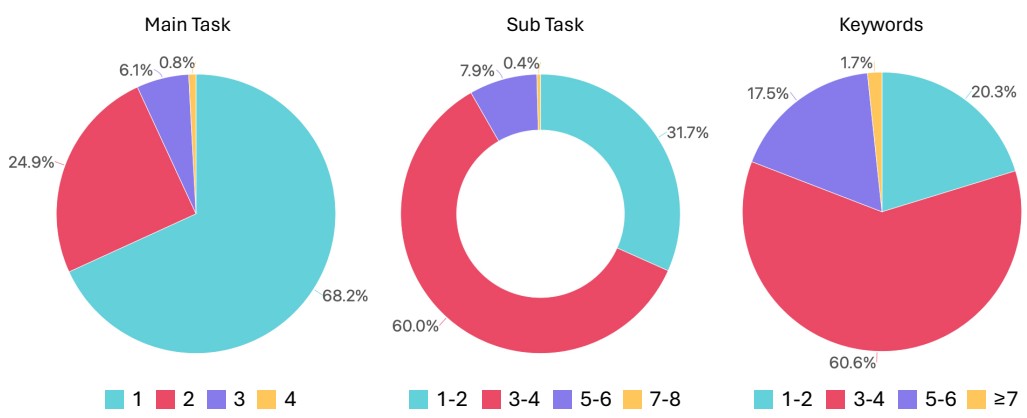

Figure 3: Distribution of number of main task, sub task and keywords.

They added missing keywords and removed extraneous ones. An initial training with 50 data points was conducted to train annotators and evaluate precision scores for maintaining quality standards.

**3. Final Scoring** After three rounds of rotational evaluation, annotators rated the revised keywords and tasks on a 5-point Likert scale, where 1 indicates 'unsatisfactory' and 5 indicates 'excellent.' For any cases with average scores lower than 4, annotators collaborated to discuss and finalize the modifications.

### 3.3 DATASET STATISTICS

Following three rounds of reviews, we finalized a dataset comprising 1,500 pairs of instructions and implementations. The dataset was partitioned into training, validation, and testing sets in a 7:2:1 ratio to ensure robust model training and evaluation.

We analyzed the distribution of the main tasks, sub-tasks, and keywords to assess the complexity of the questions. Complexity is inferred from the number of tasks a model needs to handle, which tests its reasoning and integration capabilities. Additionally, a higher count of keywords suggests a more intricate table and column setup, increasing the likelihood of errors. Figure 3 illustrates these distributions. For main tasks, 68.2% of questions involve one primary task, while 24.9% include two tasks, and 6.9% entail three or more tasks. Sub-task distribution shows that 31.7% of questions comprise one to two sub-tasks. Meanwhile, 60% involve three to four sub-tasks, and 8.3% contain over five sub-tasks. Regarding keywords, 20.3% of questions are linked to one or two keywords, 60.6% to three or four keywords, and 19.2% to five or more keywords.

## 4 THE LAIA-SQL FRAMEWORK

Current state-of-the-art methods, such as MAC-SQL Wang et al. (2023) and CHESS Talaei et al. (2024), have advanced the field of SQL generation. However, they still suffer from considerable runtime, high operation costs, and suboptimal accuracy. To address these limitations, we introduce LAIA-SQL, an innovative language-adaptive intelligent agent designed to enhance SQL generation. As shown in Figure 4, LAIA-SQL comprises three core components: User Question Understanding, Entity Retrieval, and Generation.

### 4.1 USER QUESTION UNDERSTANDING

In the initial phase of LAIA-SQL, we concentrate on thoroughly comprehending the user's question, as illustrated in Figure 4. The user's question is first incorporated into a prompt template, forming a new prompt. This prompt is then fed into a LLM to generate a response. An example of the output is displayed in Figure 4. This procedure involves two crucial tasks: *Task Decomposition* and *Keyword Extraction*:

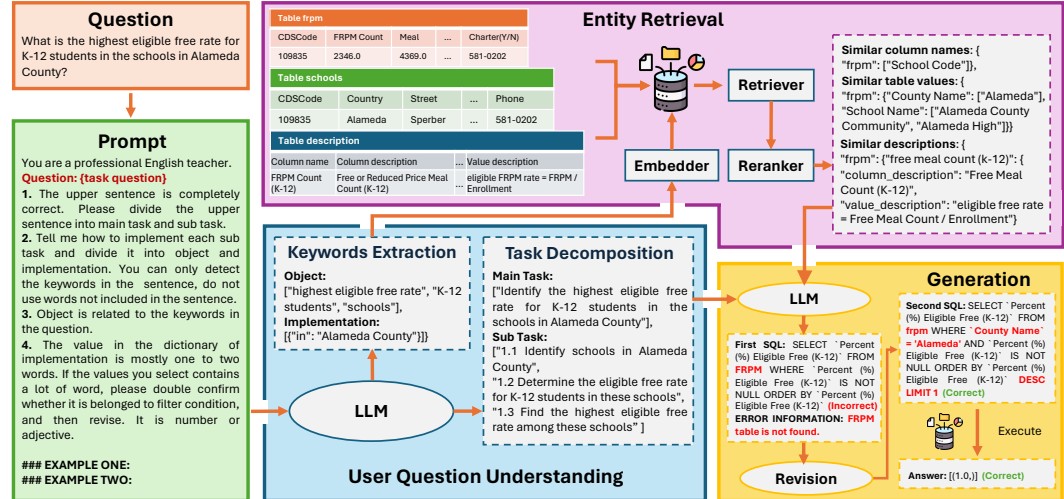

Figure 4: Framework of LAIA-SQL. Initially, the user's question is input into a prompt template, which directs LLM to perform keyword extraction and task decomposition. Keywords are then fed into the entity retrieval module to find relevant column names, table values, and descriptions. The task decomposition outcomes, entity retrieval data, and original question are then fed into the LLM, generating SQL code. If errors arise, the error information and SQL code are sent to a revision LLM for corrections. Finally, the corrected SQL code is executed to obtain the answer.

**Task Decomposition** Inspired by the COT Wei et al. (2022) reasoning approach, we decompose user questions into manageable components, addressing the inherent complexity and multi-task nature of user inquiries. Compared to previous NL2SQL methods, we employed two-level COT reasoning, which breaks down a user question into a main task and sub tasks. The main task represents the primary goal derived from the user question, while sub tasks refine main task further. This distinction aids the generation model in efficiently producing SQL code by clarifying the hierarchy of tasks. Specifically, we instruct the generation model that the main task corresponds to the main component following "SELECT," and the sub tasks correspond to operations such as "INNER JOIN," "WHERE," and "CASE WHEN," among others. However, as illustrated in Figure 2, general models like GPT-4o sometimes incorrectly decompose tasks or generate irrelevant tasks, demonstrating unstable performance. To enhance stability and reliability, we employed supervised fine-tuning for consistent task decomposition.

**Keyword Extraction** Prior methods involved merely breaking down sentences into individual keywords, which often resulted in irrelevant keywords. In our approach, we have classified keywords into two distinct categories: object and implementation, improving the accuracy. The object category encompasses terms associated with table and column names found in the user's query, whereas implementation pertains to filtering criteria, represented by a dictionary where the keys indicate filtering actions and the values denote the specified conditions. To enhance the accuracy of keyword extraction, we employed In-Context Learning (ICL) techniques to provide the LLM with multiple examples. However, as illustrated in Figure 2, GPT-4o tends to generate irrelevant or excessive keywords. To address this issue, we fine-tuned the smaller model like Mistral-7B Jiang et al. (2023) using LAIA-NLU, ultimately enhancing the accuracy of keyword extraction.

## 4.2 ENTITY RETRIEVAL

After extracting the keywords, the subsequent step involves retrieving the corresponding database entities, including table names, column names, table values, and textual descriptions (column and value descriptions). The entity retrieval component is composed of three modules: the embedder, retriever, and reranker. Initially, all table data are encoded and stored in the Chroma database. The embedder first encodes the keywords obtained during the user question understanding phase. This encoded information is then fed into the retriever to search the relevant database, yielding five entities that resemble the keywords. These five entities are subsequently passed to the reranker,

which recalculates similarity scores and reorders them, ultimately selecting the two most similar entities. Based on these three modules, we divide entity retrieval into two tasks: *Database Retrieval* and *Textual Description Retrieval*.

**Database Retrieval** In this task, our objective is to retrieve column names and table values from the database using the keywords. A column name refers to the name designated to each column, and table values are the data contained in each cell of the table, excluding the column names. To expedite the retrieval process given the extensive volume of database values, we employ two methods: Min-Hash Zhu et al. (2016) + Jaccard Score *(Equation 1 and 2)* and BM25 Robertson et al. (2009). For column names, no similarity score threshold is established during retrieval; all scores greater than 0 are recorded, and the top five highest-scoring entities are selected. For table values, if the keyword is purely numeric, we set a rule that only entities exactly matching the keyword are considered. For keywords comprising both text and numbers, no threshold is applied, and the top five highest scoring entities are selected. These shortlisted entities are then fed back into the reranker for re-ranking to identify the two most similar entities. As illustrated in Figure 4, the retrieved entities are cross-referenced to obtain their corresponding table and column names, which are then deduplicated and categorized.

$$\text{MinHash}(A, B) = \Pr(h(A) = h(B)) \tag{1}$$

$$J(A, B) = \frac{|A \cap B|}{|A \cup B|} \tag{2}$$

**Textual Description Retrieval** Textual descriptions encompass two types of information: column descriptions and value descriptions. Column descriptions provide additional details about the column names, whereas value descriptions explain the data within the columns, such as how these values were derived. Given the smaller dataset in this task, the retrieval method differs from that used in database retrieval. We directly employ an embedding model to encode the data and then use cosine similarity within the retriever to calculate scores, identifying the top five most similar entities. These entities are subsequently re-ranked using a specialized reranker model to determine the final order of relevance.

## 4.3 GENERATION

The generation process in LAIA-SQL involves two phases: *SQL Generation* and *Revision*.

**SQL Generation**: Using ICL, we guide general LLMs, like GPT-4o OpenAI (2024c), to generate SQL statements. The prompts for this task are meticulously structured into four segments: data schema, user question reasoning, constraints, and incentives. The *data schema* component includes details such as data formats, column names, table names, and examples, integrating the entity information retrieved in the Entity Retrieval module. The *user question reasoning* segment incorporates the user's question, main and sub tasks identified in User Question Understanding module, and hints derived from the dataset. By compiling these details into the prompt, the model produces an initial SQL statement.

**Revision**: As illustrated in Figure 4, the initial SQL statements may include errors such as incorrect table names, misaligned columns, or extraneous symbols. To rectify these issues, we feed the erroneous SQL statements along with their corresponding error messages back into the LLM for revision. This iterative process results in syntactically correct and operational SQL queries, ultimately yielding the correct answers.

## 5 EXPERIMENT

To rigorously assess the LAIA-SQL, we conducted a series of comprehensive experiments. These included a comparison of SQL generation accuracy on the Bird and Spider datasets against SOTA NLSQL methods. Additionally, we assessed the practical utility of LAIA-SQL against leading open-source NL2SQL methods. We also conducted ablation studies to examine the contribution of different models and modules within the LAIA-SQL. Furthermore, we evaluated the performance of

Table 1: Performance on Bird and Spider Datasets. Results from the official leaderboard.

| Method | BIRD Dataset | Spider Dataset | |
|---|---|---|---|
| | Dev EX | Dev EX | Test EX |
| GPT-4 | 46.35 | 74.0 | 67.4 |
| Distillery | 67.21 | - | - |
| CHESS | 65.00 | 87.2 | - |
| DailSQL | 54.76 | 84.4 | 86.6 |
| SFT CodeS-15B | 58.47 | 84.9 | 79.4 |
| MAC-SQL | 57.56 | 86.7 | 82.8 |
| **LAIA-SQL(ours)** | **67.28** | **88.7** | **87.1** |

Table 2: Practical utility metrics for NL2SQL methods using GPT-4o as base model.

| Method | Time(s) | Accuracy | Cost (USD) |
|---|---|---|---|
| CHESS | 119.38 | 0.5 | 11 |
| TA-SQL | 57.92 | 0.5 | 0.41 |
| SFT CodeS-15B | **35** | 0.4 | - |
| MAC-SQL | 133.55 | 0.7 | 0.38 |
| Chat2Query | 680.96 | 0.6 | - |
| **LAIA-SQL (ours)** | 56.81 | **0.8** | **0.32** |

various models fine-tuned using the LAIA-NLU dataset from multiple perspectives. Collectively, these experiments provide a multifaceted evaluation of LAIA-SQL's effectiveness.

## 5.1 EXPERIMENT SETTING

**NL2SQL Baseline Selection** We selected NL2SQL methods that are either open-source or have published papers, including GPT-4 as the baseline model. Our chosen methods are as follows: Distillery Maamari et al. (2024), which employs a schema linking augmentation technique; CHESS Talaei et al. (2024), which integrates data catalogs and database values for SQL generation; MAC-SQL Wang et al. (2023), featuring a multi-agent collaborative framework; Dail-SQL Gao et al. (2023), which combines prompt engineering with question representation, example selection, and organization; and CodeS-15B Li et al. (2024b), which uses an incremental pre-training approach on a curated SQL-centric corpus.

**Base Model Selection** In the user question understanding module, we evaluated various models such as GPT-4o-mini OpenAI (2024a), GPT-4 Achiam et al. (2023), Mistral-7B Jiang et al. (2023), LLaMA3-8B Dubey et al. (2024), Baichuan2-7B, and 13B Yang et al. (2023). For the entity retrieval module, we compared the performance of MinHash Zhu et al. (2016) combined with the Jaccard Score against BM25 Robertson et al. (2009) for the retriever. As for the embedding models, we assessed text-embedding-3-large OpenAI (2024b), Stella-1.5B, and Stella-400M. During the fine-tuning stage of the code generation model, we tested DeepSeek-Coder-V2-Instruct, DeepSeek-Coder-V2-Base Zhu et al. (2024), and Qwen-1.5-Coder Yang et al. (2024a).

**Fine-tuning Process** The fine-tuning was conducted using a setup of 4 Nvidia 4090 GPUs and utilized Distributed Data Parallel along with DeepSeed. We maintained a uniform batch size of 1 and set the epoch count to 1. The learning rate was fixed at 2e-4. Additionally, we utilized the Low-Rank Adaptation (LoRA) Hu et al. (2021) technique with specific parameters: a LoRA rank of 64, LoRA alpha of 16, and a dropout probability of 0.05. The bit precision was set to 4. It took around 30 minutes to fine-tune a LAIA-NLUer model and 4 5 hours for a code generation model.

## 5.2 METRICS

**BLEU, ROUGE and GPT-4o Score** In the evaluation of task decomposition in NLU, we assessed the quality of the generated reasoning results against human-labeled ground truth result using BLEU, ROUGE, and GPT-4o scores. Specifically, BLEU-1 and BLEU-2 provide insight into the linguistic accuracy by measuring n-gram matches between generated descriptions and ground truth Papineni et al. (2002). ROUGE-1, ROUGE-2, and ROUGE-L evaluate the overlap of n-grams, sequences, and pairs of words, offering a measure of the descriptions' comprehensiveness and relevance Lin (2004). Additionally, a five-point Likert scale evaluation by GPT-4o helps gauge the overall quality and similarity to human annotations Zheng et al. (2023).

**F1 Score** For keyword extraction tasks in NLU, the model's performance was evaluated using precision, recall, and finally get the F1 score. These metrics provide a balance between the correctness of the extracted keywords and the model's recall capability, thereby offering a holistic view of its extraction efficiency.

**Execution Accuracy (EX)** Execution accuracy was used to measure the correctness of SQL queries by comparing the results of executed predicted queries against reference queries on specific database instances. This metric not only ensures the semantic correctness but also accounts for variations in SQL formulations that yield the same results.

Table 3: Module ablation study of LAIA-SQL on dev set of Bird Dataset.

| Method | Dev EX |
|---|---|
| UQU + Entity Retrieval + Revision + Generaton(GPT-4o) | 67.28 |
| Entity Retrieval + Revision + Generaton(GPT-4o) | 59.62 |
| Entity Retrieval + Revision + Generaton | 55.28 |
| Entity Retrieval + Generaton(GPT-4) | 51.25 |
| Generaton(GPT-4) | 46.35 |

Table 4: Model ablation study of LAIA-SQL on dev set of Bird Dataset.

| Method | Dev EX |
|---|---|
| GPT-4o-mini (finetuned) + MinHASH + Stella-400M + GPT-4o | 67.28 |
| Mistral-7B (finetuned) + MinHASH + Stella-400M + GPT-4o | 65.16 |
| GPT-4 + MinHASH + Stella-400M + GPT-4o | 59.62 |
| GPT-4 + MinHASH + Stella-400M + DeepSeek-Coder-V2-Instruct (finetuned) | 55.78 |
| GPT-4 + MinHASH + Stella-400M + DeepSeek-Coder-V2-Base (finetuned) | 50.41 |
| GPT-4 + MinHASH + Stella-400M + GPT-4 | 53.17 |
| GPT-4 + MinHASH + Stella-1.5B + GPT-4 | 51.36 |
| GPT-4 + MinHASH + text-embedding-3-large + GPT-4 | 51.25 |
| GPT-4 + BM25 + text-embedding-3-large + GPT-4 | 49.34 |

## 5.3 RESULT

**BIRD and Spider Dataset Evaluation** In the BIRD dataset, due to the anonymity policy, we only report the execution accuracy on the development dataset. In the future, we will supplement with the scores for the test EX and VES. As shown in Table 1, LAIA-SQL earns the best Dev EX compared to other state-of-the-art models and is also currently the best open-source method available. In the Spider dataset, compared to all other state-of-the-art models, LAIA-SQL exhibits the highest execution accuracy across both the development and test datasets.

Additionally, in terms of practical value assessment in Table 2, we found that LAIA-SQL performs the best in aspects such as time efficiency, operational cost, and accuracy. Compared to the best open-source method CHESS, LAIA-SQL achieves a 52.4% reduction in runtime, and a 97% decrease in operational costs, demonstrating significant industrial application potential. Overall, LAIA-SQL is indeed the top-performing method among open-source NL2SQL methods.

**Ablation Study** As shown in Table 3, in our module ablation study, we observed significant improvements in accuracy with each additional module. Notably, the LAIA-NLUer, designed for keyword extraction and task decomposition, achieved the highest accuracy increase, improving by 7.66 percentage points compared to previous methods. The entity retrieval module also showed substantial gains, increasing accuracy by 4.9 percentage points. Overall, the LAIA-NLUer, entity retrieval, and revision modules are indispensable, each contributing to the improvement in accuracy.

For the result of model ablation study illustrated in Table 4, we found that within the entity retrieval module, MinHash outperformed BM25, achieving two percentage points higher accuracy and consuming only one-third of the time taken by BM25. Additionally, we observed varying performances across different embedding models. Surprisingly, the stell-400M model outperformed the stella-1.5B model, leading us to conclude that larger parameter models do not necessarily yield better embedding results.

In the code generation module, we compared the base model with fine-tuned versions and found that the fine-tuned models did not perform as well as GPT-4. However, it is important to note that our selected model only had 22 billion parameters, suggesting that the number of parameters significantly impacts the accuracy of models on complex tasks like code generation.

**Supervised fine-tuning** As shown in Table 5, we discovered that large models and small models are suited for different fine-tuning tasks. For instance, large models such as GPT-4 and GPT-4o-mini exhibit significantly better performance on complex tasks like task decomposition after fine-tuning compared to smaller models. However, for tasks that do not require deep understanding, such as keyword extraction, smaller models like Mistral-7B outperform the larger ones. Overall, our findings suggest that the decision to use large or small models for fine-tuning should be guided by

Table 5: Comparison of fine-tuned model in task decomposition and keyword extraction

| Method | BLEU | ROUGE | GPT-4o | F1 Score |
|---|---|---|---|---|
| Llama3-8B | 0.679 | 0.813 | 4.141 | 0.677 |
| Baichuan2-7B | 0.616 | 0.697 | 4.112 | 0.511 |
| Baichuan2-13B | 0.622 | 0.722 | 4.124 | 0.583 |
| Mistral-7B | 0.706 | 0.798 | 4.081 | **0.696** |
| GPT-4o-mini | 0.713 | 0.811 | 4.256 | 0.672 |
| GPT-4 | **0.722** | **0.816** | **4.286** | 0.665 |

Table 6: Impact of dataset size and epoch on the performance of LAIA-NLU on F1 Score

| Method | Dataset Size | | | | | Epoch | | | Base |
|---|---|---|---|---|---|---|---|---|---|
| | 20% | 40% | 60% | 80% | 100% | 1 | 2 | 3 | Model |
| Llama3-8B | 0.609 | 0.636 | **0.677** | 0.661 | 0.653 | 0.677 | 0.728 | 0.734 | 0.442 |
| Baichuan2-7B | 0.497 | 0.515 | 0.558 | 0.522 | 0.511 | 0.511 | 0.648 | 0.688 | 0.208 |
| Mistral-7B | **0.648** | **0.640** | 0.634 | **0.694** | **0.696** | **0.696** | **0.755** | **0.769** | **0.502** |
| Baichuan2-13B | 0.412 | 0.554 | 0.573 | 0.638 | 0.585 | 0.585 | 0.609 | 0.647 | 0.266 |

the specific requirements of the task, as the performance of fine-tuned large models is not universally superior.

In addition, we compared the effects of varying dataset sizes and different epochs on fine-tuning performance on keyword extraction. In Table 6, we found that the overall performance of the Mistral-7B model was the best, followed by the LLaMA-8B model. Notably, we observed that for all models except Mistral-7B, the F1-Score initially increased and then decreased as the training data size increased. This indicates that more data is not always better. Moreover, we discovered that increasing the number of epochs significantly improved the F1-Score, suggesting that adding more epochs is the most effective method for enhancing the accuracy of keyword extraction.

## 6 LIMITATION

While our model surpasses many state-of-the-art NL2SQL methods, its accuracy still falls short for practical use. Fine-tuning on specific datasets is essential for satisfactory performance, highlighting the need for enhanced generalizability across varied domains. Computational limitations confined us to training smaller models; larger models like DeepSeek-V2-Coder-236B and Llama3.1-70B could potentially offer superior performance over our current 22B model, thereby significantly improving accuracy. Additionally, the Entity Retrieval component of LAIA-SQL employs MinHash with Jaccard Score and BM25, resulting in suboptimal retrieval performance. Leveraging advanced RAG modules could enhance this aspect. Furthermore, LAIA-NLU dataset is limited to 1500 samples due to resource constraints, affecting the LAIA-NLUer model's robustness. The scarcity of high-quality data, exacerbated by copyright restrictions, presents a significant challenge. Future work should prioritize data augmentation techniques and innovative methods to mitigate data scarcity, as well as improving computational resources to explore more advanced models.

## 7 CONCLUSION

In this work, we introduced significant advancements in Table QA methods by developing the LAIA-NLU dataset and a retrieval-augumented based NL2SQL framework, LAIA-SQL. Our meticulously curated dataset, containing 1,500 high-quality instructions, enabled us to train LAIA-NLUer, a pioneering NLU model tailored for Table QA. By integrating LAIA-NLUer, our NL2SQL method LAIA-SQL demonstrated remarkable improvements, achieving higher accuracy to 67.28% and reducing SQL query execution time by 52.4% to 56.81 second for 10 questions. Meanwhile, the cost is reduced to 0.032 USD for one question. These findings underscore the potential of our approach to enhance the efficiency and accuracy of multi-table data retrieval, making it more accessible to non-expert users.

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

# A  APPENDIX

You are a professional English teacher.
**Question: {task question}**
**1.** The upper sentence is completely correct. Please divide the upper sentence into main task and sub task.
**2.** Tell me how to implement each sub task and divide it into object and implementation. You can only detect the keywords in the  sentence, do not use words not included in the sentence.
**3.** Object is related to the keywords in the question.
**4.** The value in the dictionary of implementation is mostly one to two words. If the values you select contains a lot of word, please double confirm whether it is belonged to filter condition, and then revise. It is number or adjective.
**5.** Please only respond with a JSON object structured as follows, don't change the keys name.

**### EXAMPLE ONE:**
{
'question':"Name schools in Riverside which the average of average math score for SAT is grater than 400, what is the funding type of these schools?",
'main task':["1. Name schools in Riverside which the average of average math score for SAT is grater than 400", "2. what is the funding type of these schools?"],
'sub task':["1.1 find the name of schools in Riverside",
"1.2 get the average math score of these school",
"1.3 calculate the average score of average math score of eah school.",
"1.4 find the school which the average of average math score for SAT is grater than 400",
"2.1 the funding type of these schools"],
'object':['Name schools','funding type', 'average math score for SAT','schools'],
'implementation':[{'in':'Riverside'}, {'is grater than':'400'}]
}

**### EXAMPLE TWO:**
{
'question': "How many units of item no.9 were sold in store no.1 in total in January, 2012?",
' main task': ["Determine the total units sold of item no.9 in store no.1 in January, 2012"],
'sub task': ["1.1 Identify store no.1",
"1.2 Identify item no.9",
"1.3 Track sales in January, 2012",
"1.4 Calculate total units sold of item no.9"],
'object': ['units', 'item no', 'store no'],
'implementation': [{'store no.': '1'}, {'item no.': '9'}, {'in': 'January, 2012'}]
}

Figure 5: Prompt of keyword extraction and task decomposition.

You are a data science expert.
Below, you are presented with a database schema and a question.
Your task is to read the schema, understand the question, and generate a valid SQLite query to answer the question.
Before generating the final SQL query think step by step on how to write the query.

**### Database Schema**
**{DATABASE_SCHEMA}**

This schema offers an in-depth description of the database's architecture, detailing tables, columns, primary keys, foreign keys, and any pertinent information regarding relationships or constraints.
Pay attention!!! Special attention should be given to the examples listed beside each column of data schema, as they directly hint at which columns are relevant to our query.

**### Constraints**
1. For key phrases mentioned in the question, we have provided the most similar values within the columns denoted by "-- examples" in front of the corresponding column names. This is a crucial hint indicating the correct columns to use for your SQL query.
2. pay attention!!! avoid using different column for the same object with different filter values.
3. pay attention!!! Don't write a wrong column in the SQL code. Please check whether the column is belong to the table again in the SQL.

**### Question:**
**{QUESTION}**

**### Steps that you should follow:**
**{Main Task}**
**{Sub Task}**
**{Hint}**

The main task, sub task and evidence are correct, please base on them generate final sql query, please strictly follow the main task, sub task and evidence.
If there is an equation in the evidence, please strictly follow the equation!!!
The amount of item SELECT in sql query depends on the number of main tasks. if there is only one main task, you should only SELECT one item related to the main task in the sql query.

Please respond with a JSON object structured as follows:
{{"SQL": "Your SQL query is here."}}

Figure 6: Prompt of candidate generation.

Objective: Your objective is to make sure a query follows the database admin instructions and use the correct conditions.

**Database Schema:**
**{DATABASE_SCHEMA}**

**### Constraints**
1. When you need to find the highest or lowest values based on a certain condition, using ORDER BY + LIMIT 1 is prefered over using MAX/MIN within sub queries.
2. If predicted query includes an ORDER BY clause to sort the results, you should only include the column(s) used for sorting in the SELECT clause if the question specifically ask for them. Otherwise, omit these columns from the SELECT.
3. Predicted query should return all of the information asked in the question without any missing or extra information.
4. For key phrases mentioned in the question, we have provided the most similar values within the columns denoted by "-- examples" in front of the corresponding column names. This is a crucial hint indicating the correct columns to use for your SQL query.
5. If you are joining multiple tables, make sure to use alias names for the tables and use the alias names to reference the columns in the query. Use T1, T2, T3, ... as alias names.

**### Question:**
**{QUESTION}**

**### ERROR INFORMATION**
**{Error Infomation}**

**### Steps that you should follow:**
**{Main Task}**
**{Sub Task}**
**{Hint}**

**### Predicted query:**
**{SQL}**

Pay attention to the ERROR INFORMATION, based on the error revise the SQL query.
Think about whether the predicted query used the hint and evidence already, if not, use the hint and evidence in the sql query generation.

Please respond with a JSON object structured as follows (if the sql query is correct, return the query as it is):
{{"revised_SQL": "Your revised SQL query is here."}}

Figure 7: Prompt of revision.

