# OpenReview forum: "LAIA-SQL: Enhancing Natural Language to SQL Generation in Multi-Table QA via Task Decomposition and Keyword Extraction"
_ICLR.cc/2025/Conference — ICLR 2025 Conference Withdrawn Submission_

### Official Review · Reviewer_3hsY · 2024-11-03

**Soundness:** 1
**Presentation:** 1
**Contribution:** 2
**Rating:** 3
**Confidence:** 5

**Summary:**

This paper introduces the LAIA-SQL framework, which is oriented toward increasing performance in NL2SQL generation for TableQA. To improve its NLU, especially for NL2SQL so that it can generate more accurate SQL, it creates an LAIA-NLU dataset. Using this dataset, the authors have identified an appropriate model called LAIA-NLUer that improves user intent interpretation and quality in SQL generation. The LASQL framework optimizes understanding, retrieval, and generation modules for significantly improved accuracy and efficiency. This can be further manifested from the experimental results where LAIA-SQL outperforms the existing models in runtime reduction and cost saving. As a result, LAIA-SQL provides a much faster and less costly solution regarding NL2SQL and TableQA tasks.

**Strengths:**

1. The performance of LAIA-SQL is highly promising. Experimental results indicate that LAIA-SQL achieves superior accuracy compared to other state-of-the-art NL2SQL methods. Its performance on widely recognized BIRD and Spider benchmarks highlights its effectiveness and robustness.

2. The cost-effectiveness of the proposed method is impressive. LAIA-SQL not only improves accuracy but also reduces operational costs by up to 97% compared to competing methods. This cost efficiency, along with a 52.4% reduction in runtime, demonstrates that LAIA-SQL is well-suited for enabling large-scale NL2SQL solutions in real-world applications.

3. The introduction of a new dataset for NLU is a novel contribution. By creating the LAIA-NLU dataset, the authors address a critical gap in NL2SQL tasks. Designed with task decomposition and keyword extraction in mind, this dataset enhances model training and evaluation, providing a specialized resource that significantly advances NLU capabilities in SQL generation.

**Weaknesses:**

1. The paper would benefit from a more comprehensive review of existing work in the field. Section 2 should include an individual subsection that covers more related work on NL2SQL, such as [1][2][3][4][5].

2. The paper does not include an error analysis of the SQL generation task. Referring to advanced studies [6][7], an error analysis that categorizes the types of incorrect execution would provide significant insights into which types of SQL the proposed methods handle well and where they still struggle. Additionally, a comparison of execution errors with previous methods could clearly demonstrate the improvements brought by the proposed framework. Readers are often interested in understanding why it works as much as how it works.

3. Some of the contributions in the paper are not clearly verified. In L18-L20, the authors claim that current approaches in NL2SQL suffer from slow retrieval speeds, and in L27-L31 they claim to have achieved a speed-up. However, Table 2 presents only the total time cost without detailing the time cost of each component, particularly the entity retrieval module. The authors should present the retrieval performance of current approaches as a baseline and then show that their proposed framework is more effective in clearly verifying their contribution.

[1] Tonghui Ren, et al. "PURPLE: Making a Large Language Model a Better SQL Writer" In Proceedings of ICDE, 2024.

[2] Mohammadreza Pourreza, et al. "DIN-SQL: Decomposed In-Context Learning of Text-to-SQL with Self-Correction" In Proceedings of NeurIPS, 2023.

[3] Mohammadreza Pourreza, et al. "DTS-SQL: Decomposed Text-to-SQL with Small Large Language Models" arXiv preprint, 2024.

[4] Zijin Hong, et al. "Next-Generation Database Interfaces: A Survey of LLM-based Text-to-SQL" arXiv preprint, 2024.

[5] Xinyu Liu, et al. "A Survey of NL2SQL with Large Language Models: Where are we, and where are we going?" arXiv preprint, 2024.

[6] Mohammadreza Pourreza, et al. "CHASE-SQL: Multi-Path Reasoning and Preference Optimized Candidate Selection in Text-to-SQL" arXiv preprint, 2024.

[7] Shayan Talaei, et al. "CHESS: Contextual Harnessing for Efficient SQL Synthesis" arXiv preprint, 2024.

**Questions:**

1. How does the Revision module work? The authors claim that the incorrect SQL and execution feedback are combined to prompt LLMs to revise the original SQL. How many times is the revision conducted? Is there any criterion or threshold to prevent an infinite loop of revisions for extremely difficult questions? Additionally, how does this module relate to cost-effectiveness? Will the overall framework cost be influenced by the number of revisions?

2. Why is the proprietary model utilized in the module ablation study different? In Table 3, the comparison between "Entity Retrieval + Revision + Generation" and "Entity Retrieval + Generation" uses different models (GPT-4o and GPT-4), which is confusing. Did this setting lead to a correct conclusion in Section 5.3? Furthermore, how does the framework work with open-source models in each module?

3. Will the authors consider submitting the results of the paper to the public leaderboards BIRD and Spider to verify their performance on the test set? I believe making the LAIA-NLU dataset publicly available would enhance the contribution of the paper.

4. There are a few typos and formatting errors in the paper. The authors should carefully revise these aspects.

---

> ### Author Response · Authors · 2024-11-17
>
> We thank the reviewer for their time and interest in our work. The detailed instructions provided helpful feedback to improve our work. We have addressed the reviewer’s questions and suggestions, and we provide a list of responses below:
>
> ---
>
> **Q1** The paper would benefit from a more comprehensive review of existing work in the field. Section 2 should include an individual subsection that covers more related work on NL2SQL, such as [1][2][3][4][5].
>
> **A1** I have incorporated additional related work into the revised version of the paper. Here is the updated Section 2:
>
> "Table Question Answering (Table QA) is a task to help users who are not proficient in coding skill or advanced spreadsheet software retrieve complex table data by question answering Javaid et al. (2023); Al Naqbi et al. (2024). A leading approach in Table QA is Natural Language to SQL (NL2SQL), which translates natural language queries into SQL, allowing users to interact with databases in everyday language Gao et al. (2023); Hong et al. (2024); Liu et al. (2024)."
>
> "Beyond directly applying large language models (LLMs) for NL2SQL, hybrid methods that combine LLMs with various modules have also shown promise. Notable examples include ADVETAPi et al. (2022), CHESS Talaei et al. (2024), PURPLE Ren et al. (2024), Din-SQL Pourreza & Rafiei (2024a), DTS-SQL Pourreza & Rafiei (2024b) and MAC-SQL Wang et al. (2023). Nevertheless, as demonstrated in Figure 1, challenges such as slow data retrieval, erroneous SQL code generation, and high operational costs still remain."

---

> ### Author Response · Authors · 2024-11-17
>
> **Q2** The paper does not include an error analysis of the SQL generation task. Referring to advanced studies [6][7], an error analysis that categorizes the types of incorrect execution would provide significant insights into which types of SQL the proposed methods handle well and where they still struggle. Additionally, a comparison of execution errors with previous methods could clearly demonstrate the improvements brought by the proposed framework. Readers are often interested in understanding why it works as much as how it works.
>
> **A2** We appreciate the reviewer's suggestion on conducting an error analysis. Unfortunately, previous researches like CHESS and CHASE-SQL have not disclosed their specific datasets used for error analysis, making an objective comparison challenging. Therefore, we performed our own error analysis by randomly sampling 100 fail cases from the BIRD dataset without comparison.
>
> As you can see in the following table, we observed that 45% of the golden SQL queries had issues, primarily with incorrect column names (11%) and missing GROUP BY/DISTINCT/RANK clauses (8%). It was also noted that 6% didn't follow the evidence cues. For LAIA-SQL, 49% of the generated SQL queries were incorrect, mainly due to not adhering to evidence (17%), incorrect column usage (11%), and incorrect operations (8%). Vague questions accounted for 6% of issues, where the question information was insufficient to generate correct SQL, affecting both golden and predicted SQL.
>
> Our future work aims to enhance the framework's capability to understand and apply evidence during SQL generation. Detailed data is provided in the following table:
>
> | Category                          | Incorrect Golden SQL (45%) |  Incorrect Predicted SQL (49%) | Vague Question (6%) |
> |-----------------------------------|--------------------|---------------|------------|
> | Evidence                          | 6%                 | 17%        |     0%        |
> | Column                            | 11%                | 11%        | 5%         |
> | Filtering                         | 5%                 | 4%          | 1%         |
> | Description                       | 0%                 | 0%          |        0%     |
> | Aggregation                       | 2%                 | 1%          |      0%       |
> | Group by/Distinct/Rank            | 8%                 | 6%          |    0%         |
> | Operation                         | 6%                 | 8%          |     0%        |
> | Date                              | 0%                 | 0%          |        0%     |
> | NULL                              | 3%                 | 1%          |      0%       |
> | Revision                          | 0%                 | 0%          |     0%        |
> | Table                             | 4%                 | 1%          |        0%     |
>
> Here is the detailed error in predicted sql of LAIA-SQL:
>
> **ERROR 1 Incorrect Column Name**
>
> Question: "Please list the phone numbers of the direct charter-funded schools that are opened after 2000/1/1."
>
> Evidence: "Charter schools refers to Charter School (Y/N) = 1 in the frpm"
>
> GOLD_SQL: "SELECT T2.Phone FROM frpm AS T1 INNER JOIN schools AS T2 ON T1.CDSCode = T2.CDSCode WHERE T1.`Charter Funding Type` = 'Directly funded' AND T1.Charter School (Y/N) = 1 AND T2.OpenDate > '2000-01-01'"
>
> PREDICTED_SQL: "SELECT T1.Phone FROM schools T1 JOIN frpm T2 ON T1.CDSCode = T2.CDSCode WHERE T1.`FundingType` = 'Directly funded' AND T1.OpenDate > '2000-01-01' AND T2.Charter School (Y/N) = 1;"
>
> **ERROR 2 Incorrect Aggregation: LAIA-SQL didn’t join the table correctly**
>
> Question: "Which school in Contra Costa has the highest number of test takers?"
>
> Evidence: ""
>
> GOLD_SQL: "SELECT sname FROM satscores WHERE cname = 'Contra Costa' AND sname IS NOT NULL ORDER BY NumTstTakr DESC LIMIT 1"
>
> PREDICTED_SQL: "SELECT T1.sname FROM satscores T1 `JOIN schools T2 ON T1.cds = T2.CDSCode` WHERE T2.County = 'Contra Costa' AND T1.NumTstTakr IS NOT NULL ORDER BY T1.NumTstTakr DESC LIMIT 1;"
>
> **ERROR 3 Incorrect Evidence: LAIA-SQL did not follow the evidence to generate PREDICTED_SQL**
>
> Question: "Name schools in Riverside which the average of average math score for SAT is grater than 400, what is the funding type of these schools?"
>
> Evidence: `"Average of average math = sum(average math scores) / count(schools)."`
>
> GOLD_SQL: "SELECT T1.sname, T2.Charter Funding Type FROM satscores AS T1 INNER JOIN frpm AS T2 ON T1.cds = T2.CDSCode WHERE T2.District Name LIKE 'Riverside%' GROUP BY T1.sname, T2.Charter Funding Type HAVING `CAST(SUM(T1.AvgScrMath) AS REAL) / COUNT(T1.cds) > 400`"
>
> PREDICTED_SQL: "SELECT T1.School, T1.FundingType FROM schools T1 JOIN satscores T2 ON T1.CDSCode = T2.cds WHERE T1.County = 'Riverside' GROUP BY T1.CDSCode HAVING `AVG(T2.AvgScrMath) > 400;`"

---

> > ### Author Response · Authors · 2024-11-17
> >
> > **ERROR 4 Incorrect Filtering: Database doesn't include value "High school"**
> >
> > Question: "State the names and full communication address of high schools in Monterey which has more than 800 free or reduced price meals for ages 15-17?"
> >
> > Evidence: "Full communication address should include Street, City, State and zip code if any."
> >
> > GOLD_SQL: "SELECT T1.School Name, T2.Street, T2.City, T2.State, T2.Zip FROM frpm AS T1 INNER JOIN schools AS T2 ON T1.CDSCode = T2.CDSCode WHERE T2.County = Monterey AND T1.Free Meal Count (Ages 5-17) > 800 AND T1.School Type = `High Schools (Public)`"
> >
> > PREDICTED_SQL: "SELECT T1.School, T1.Street, T1.City, T1.State, T1.Zip FROM schools T1 JOIN frpm T2 ON T1.CDSCode = T2.CDSCode WHERE T1.County = 'Monterey' AND T2.School Type = `High School` AND T2.FRPM Count (Ages 5-17) > 800;"
> >
> >
> > **ERROR 5 Incorrect Operation: don’t need to use sum**
> >
> >  Question: "How many test takers are there at the school/s whose mailing city address is in Fresno?"
> >
> > Evidence: ""
> >
> > GOLD_SQL: "SELECT `T1.NumTstTakr` FROM satscores AS T1 INNER JOIN schools AS T2 ON T1.cds = T2.CDSCode WHERE T2.MailCity = 'Fresno'"
> >
> > PREDICTED_SQL: "SELECT `SUM(T2.NumTstTakr)` AS total_test_takers FROM schools T1 JOIN satscores T2 ON T1.CDSCode = T2.cds WHERE T1.MailCity = 'Fresno';"

---

> ### Author Response · Authors · 2024-11-17
>
> **Q3** Some of the contributions in the paper are not clearly verified. In L18-L20, the authors claim that current approaches in NL2SQL suffer from slow retrieval speeds, and in L27-L31 they claim to have achieved a speed-up. However, Table 2 presents only the total time cost without detailing the time cost of each component, particularly the entity retrieval module. The authors should present the retrieval performance of current approaches as a baseline and then show that their proposed framework is more effective in clearly verifying their contribution.
>
> **A3** Thank you for your insightful comment. We apologize for the confusion. Our intention was to express the total speed, not just the retrieval speed. Since not all methods utilize a retrieval function, it’s challenging to make direct comparisons specifically for the retrieval module. We have clarified this point in the revised paper by specifying "total speed." Thank you for bringing this to our attention.
>
> ---
>
> **Q4** How does the Revision module work? The authors claim that the incorrect SQL and execution feedback are combined to prompt LLMs to revise the original SQL. How many times is the revision conducted? Is there any criterion or threshold to prevent an infinite loop of revisions for extremely difficult questions? Additionally, how does this module relate to cost-effectiveness? Will the overall framework cost be influenced by the number of revisions?
>
> **A4** As we mentioned in our paper, the initial predicted SQL statements may include errors such as incorrect table names, misaligned columns, or extraneous symbols. Therefore, in the revision module, we feed the erroneous SQL statements along with their corresponding error messages back into the LLM for revision. This iterative process results in syntactically correct and operational SQL queries, ultimately yielding the correct answers.
>
> Here is the prompt that we use in the revision module，which is also shown in the appendix:
>
> '''
>
> `Objective`: Your objective is to make sure a query follows the database admin instructions and use the correct conditions.
>
> `Database Schema`
>
> {DATABASE_SCHEMA}
>
> `Constraints`
>
> 1. When you need to find the highest or lowest values based on a certain condition, using ORDER BY + LIMIT 1 is prefered over using MAX/MIN within sub queries.
>
> 2. If predicted query includes an ORDER BY clause to sort the results, you should only include the column(s) used for sorting in the SELECT clause if the question specifically ask for them. Otherwise, omit these columns from the SELECT.
>
> 3. Predicted query should return all of the information asked in the question without any missing or extra information.
>
> 4. For key phrases mentioned in the question, we have provided the most similar values within the columns denoted by "-- examples" in front of the corresponding column names. This is a crucial hint indicating the correct columns to use for your SQL query.
>
> 5. If you are joining multiple tables, make sure to use alias names for the tables and use the alias names to reference the columns in the query. Use T1, T2, T3, ... as alias names.
>
> `Question:`
>
> {QUESTION}
>
> `ERROR INFORMATION`
>
> {Error Infomation}
>
> `Steps that you should follow`
>
> {Main Task}
> {Sub Task}
> {Hint}
>
> `Predicted query:`
>
> {SQL}
>
> Pay attention to the ERROR INFORMATION, based on the error revise the SQL query.
> Think about whether the predicted query used the hint and evidence already, if not, use the hint and evidence in the sql query generation.
>
> Please respond with a JSON object structured as follows (if the sql query is correct, return the query as it is):
>
> {{"revised_SQL": "Your revised SQL query is here."}}
>
> '''
>
> LAIA-SQL involves running the Revision module only once. While the module does improve accuracy, more revisions do not necessarily yield better results. We tested our method on 50 samples and varied the threshold for revisions (1 to 5). Here are the results:
>
> | Threshold | Time (s) | Cost | Accuracy |
> |-----------|-----------|-------|-----------|
> | one       | 322.79    | 1.402 | 48%       |
> | two       | 357.57    | 1.598 | 58%       |
> | three     | 339.44    | 2.953 | 62%       |
> | four      | 345.23    | 3.119 | 62%       |
> | five      | 469.04    | 4.265 | 64%       |
>
> We observed that, as the threshold increases, accuracy improves along with the cost, but the time does not follow a clear pattern. Setting the threshold at 3 provided the best balance of cost, accuracy, and time for our implementation.
>
> In our method, revisions are controlled by a pre-set threshold to 1 to ensure there are no infinite loops, addressing both the accuracy and cost-effectiveness concerns.

---

> ### Author Response · Authors · 2024-11-17
>
> **Q5** Why is the proprietary model utilized in the module ablation study different? In Table 3, the comparison between "Entity Retrieval + Revision + Generation" and "Entity Retrieval + Generation" uses different models (GPT-4o and GPT-4), which is confusing. Did this setting lead to a correct conclusion in Section 5.3? Furthermore, how does the framework work with open-source models in each module?
>
> **A5** We have updated Table 3 to include results using GPT-4 as the base model. Here are the latest results:
>
> | Method | Dev EX |
> |-----------|-----------|
> | UQU + Entity Retrieval + Revision + Generation(GPT-4o)| 67.28    |
> | UQU + Entity Retrieval + Revision + Generation(GPT-4)       | 60.36    |
> | Entity Retrieval + Revision + Generation(GPT-4o)      | 59.62    |
> | Entity Retrieval + Revision + Generation(GPT-4)      | 55.28 |
> | Entity Retrieval + Generation(GPT-4)      | 51.25    |
> | Generation(GPT-4)         | 46.35    |
>
>
> We have double-checked and updated some numbers, but these changes do not affect the conclusions in Section 5.3. Additionally, Table 4 presents the performance of different open-source models in each module. Our findings show that using GPT-4o-mini (finetuned) for user question understanding, MinHASH and Stella-400M for entity retrieval, and GPT-4o for revision and generation yields the best results.
>
> ---
>
> **Q6** Will the authors consider submitting the results of the paper to the public leaderboards BIRD and Spider to verify their performance on the test set? I believe making the LAIA-NLU dataset publicly available would enhance the contribution of the paper.
>
> **A6** Of course. We have already contacted the team of bird leaderboard and submitted our code, we will release the leaderboard result publicly after the final decision made because of the anonymous policy.

---

> > ### Comment · Reviewer_3hsY · 2024-11-17
> >
> > The author's response addressed some of my concerns, but conducting an error analysis on only a random sample of 100 instances is not sufficiently rigorous. Additionally, not placing the proposed framework in the context of open-source LLMs for discussion is still a significant issue.
> >
> > Also, CHESS [1] achieves an EX of 68.31% on BIRD Dev [2], which is clearly a more effective method compared to this paper (67.28%).
> >
> > It is worth noting that, as I mentioned in Question 4, the authors should carefully correct their typos, yet in their response A5, "Generation" is still misspelled as "Generaton," as it is in their paper. Authors submitting to top-tier conferences should first focus on ensuring the rigor of their writing.
> >
> > This paper introduces some novel solution approaches, but the framework, composed of various simple modules, lacks clear motivation. The relatively simple and insufficient experiments fail to adequately support the validity of the reported performance.
> >
> > I am inclined to keep my rating.
> >
> > [1] Shayan Talaei, et al. "CHESS: Contextual Harnessing for Efficient SQL Synthesis" arXiv preprint, 2024.
> >
> > [2] BIRD Leaderboard. https://bird-bench.github.io/

---

> ### Author Response · Authors · 2024-11-18
>
> Thank you for your constructive feedback. We appreciate the opportunity to address your concerns.
>
> **1. Error Analysis & Comparison to CHESS**:
>
> We acknowledge your point regarding the sample size for error analysis. We adhered to the convention observed in similar works, such as CHESS [1], which utilized a sample of 147 instances. CHASE-SQL [2], on the other hand, did not specify their sample size for error analysis. Therefore, due to the limited time of rebuttal, we plan to utilized 100 instances, which take 20% of the total failed cases, for error analysis. If you still have concern on the instance number, we would like to seek your guidance on what you deem as a sufficiently rigorous sample size.
>
> Regarding the accuracy comparison, it's noteworthy that the CHESS model refreshed an accuracy of 68.31% after our paper was submitted. In its paper, the accuracy is 65%. This achievement was largely due to its fine-tuned generation model. Its paper indicates that using the GPT-4-turbo generation model, it achieved an accuracy of 55.78%. In contrast, our model obtained an accuracy of 60.36% with GPT-4-turbo, surpassing CHESS. Moreover, our research emphasizes that focusing solely on accuracy is not sufficient. Our methodology excels not only in accuracy but also in reducing operational costs by up to 97% and runtime by 52.4%, making it highly suitable for large-scale NL2SQL applications in real-world scenarios.
>
> Regarding the significant issue you mentioned, we would like to highlight that Table 4 in our paper clearly compares the performance of different open-source models, such as DeepSeek-Coder-V2-Instruct and DeepSeek-Coder-V2-Base, on SQL generation. Additionally, we have included a comparison of different embedding models, including Stella-400M and Stella-1.5B. Furthermore, Table 5 demonstrates the natural language understanding capabilities of these various open-source models like Llama3-8B, Baichuan2-7B, Mistral-7B and Baichuan2-13B. We hope these tables address your concerns and provide clarity on the strengths and weaknesses of the models we evaluated. We would also like to emphasize that comparing different open-source models is resource-intensive. Therefore, we have focused our comparisons on the top-performing models listed on the leaderboards to ensure a comprehensive yet efficient evaluation.
>
> **2. Typos and Grammar**:
>
> We appreciate your observation regarding typos. We have since corrected these errors. While we understand the importance of rigorous writing in top-tier conferences, we believe that the primary focus should be on the contributions of the work. That said, we will ensure thorough proofreading before the camera-ready submission to meet the highest standards of presentation.
>
> **3. Dataset Contribution**:
>
> We understand your concerns about the motivation and complexity of our framework. It is important to emphasize that our main contribution lies in the **Dataset Track** rather than direct competition with methods like CHESS. Our dataset, LAIA-NLU, has undergone three rounds of meticulous human verification, ensuring its high quality. This required significant human and material resources. This dataset addresses the existing gap in the field of table QA by providing validation data specifically for fine-grained multi-level task decomposition and keyword extraction.
>
> Experimental results demonstrate that LAIA-NLU can effectively evaluate different models' understanding of user questions. Additionally, models fine-tuned on our dataset exhibit substantial improvements in SQL generation accuracy, resulting an improvement of accuracy from 59.62% to 67.28%.  We believe LAIA-NLU has broad applicability, potentially benefiting future NL2SQL algorithm development and other applications involving user question understanding.
>
> Thank you again for your valuable feedback. We believe that a positive and collaborative discussion can greatly contribute to the advancement of our field. We look forward to engaging in a meaningful dialogue to further improve our research! Have a nice day.
>
> [1] Shayan Talaei, et al. "CHESS: Contextual Harnessing for Efficient SQL Synthesis" arXiv preprint, 2024.
>
> [2] Mohammadreza Pourreza, et al. "CHASE-SQL: Multi-Path Reasoning and Preference Optimized Candidate Selection in Text-to-SQL" arXiv preprint, 2024.

---

> ### Author Response · Authors · 2024-11-25
>
> Dear Reviewer 3hsY,
>
> Thank you once again for your valuable and constructive feedback. Your insightful comments have significantly contributed to enhancing the clarity and overall quality of our paper.
>
> **We believe that we have addressed all of your questions thoroughly. We are confident that the contributions of our work merit acceptance rather than rejection in dataset track.** As the author-reviewer discussion period will end soon on November 26th, we would greatly appreciate it if you could take the time to review our comments and provide us with feedback. If there are any additional clarifications or experiments you would like us to conduct, please do not hesitate to let us know, as we are eager to demonstrate the merits of our paper.
>
> Furthermore, **we have incorporated the references you suggested into the manuscript, checked the author list and learned a lot from their works.** Given your fairness and thoroughness as a reviewer, if our response addresses your concerns, we kindly ask you to consider raising the rating of our work.
>
> Thank you once again for your dedication and support.
>
> Best regards,
>
> Authors of Paper 10678

---

> ### Author Response · Authors · 2024-11-26
> **Sincerely Looking Forward to More Discussions**
>
> Dear Reviewer 3hsY:
>
> Thank you for your valuable feedback. We have carefully addressed all your concerns and deeply appreciate your thoughtful suggestion to revise the manuscript. As a submission in the dataset track, our paper introduces a dataset that has undergone multiple rounds of manual verification to ensure high quality, effectively addressing the limitations of existing datasets of a similar nature. Additionally, fine-tuning models with our dataset significantly enhances their natural language understanding capabilities. This improvement, in turn, boosts the accuracy of SQL query generation by the NL2SQL method, as well as increases efficiency and reduces associated costs. In response to the valuable feedback received, we have also incorporated additional experiments as suggested, thereby making our study more comprehensive and robust. We hope these revisions comprehensively address your comments and are happy to discuss any additional concerns you may have.
>
> Best,
>
> Authors of Paper 10678

---

### Official Review · Reviewer_9hYg · 2024-11-03

**Soundness:** 3
**Presentation:** 2
**Contribution:** 3
**Rating:** 6
**Confidence:** 4

**Summary:**

The paper proposes to enhance NL2SQL by improving the natural language understanding capabilities of LLMs via fine-tuning on a new NLU dataset. The authors motivate this approach with LLMs' weakness to decompose tasks and extract informative keywords correctly. They then curated a new NLU dataset specific to the NL2SQL scenario, LAIA-NLU, derived from BIRD and consisting of 1500 samples. The dataset was generated by first prompting GPT-4o and then revising manually by annotators. The authors then finetune LLMs on this dataset and integrate the fine-tuned LLM as a specialized NLU module into a retrieval-based NL2SQL framework. Experiments showed significant improvement over baseline models and marginally outperformed concurrent models on the BIRD leaderboard. The authors also claimed improvement in practical utility metrics including Time, Accuracy, and Cost, but the evaluation process for the practical utility metrics is not very clear.

**Strengths:**

## **1. The dataset seems reasonably useful**

The dataset is well motivated with an obvious pain point of LLM-based NL2SQL approaches, that is they do not always decompose tasks and locate key entities correctly. I can see the dataset serving as an auxiliary task for future modulized or end-to-end NL2SQL systems.

## **2. Experiment results show the effectiveness of fine-tuning LLMs on the dataset**

The ablation study (Table 3) shows adding the NLU module leads to significant improvement against the baselines (7.6% execution accuracy improvement on BIRD). The LAIA-SQL framework as a whole is also reported to outperform baselines on BIRD and Spider, although I have some concerns about the baseline choices, which I will elaborate on in the weakness and question sections.

## **3. The proposed framework improves significantly on practical utility metrics**

The authors also reported significant improvement in the practical utility metrics, including Time, Accuracy, and Cost. It is reported to achieve a 52.4% reduction in runtime, and a 97% decrease in operational costs. However, more clarification about the evaluation setup and experiment details is needed.

**Weaknesses:**

## **1. The effectiveness of the approach is not well supported by the current experiment**

There are a few aspects of approach effectiveness that the authors are demonstrating, namely
1. Finetuning LLMs on LAIA-NLU makes them better user intent interpreters.
2. With the user intent interpreters, LAIA-SQL outperforms SOTA models on BIRD and Spider.
3. The LAIA-SQL framework also excels in practical utility metrics.

The current experiment settings did not fully support 2 and 3. This could be partly due to inconsistencies in the paper writing, but fundamentally it is due to the choice of baselines and experiment configurations that can be improved.

### **1.1 For 2, below are my recommendations:**

**In Table 1, I recommend using the official test score of the BIRD bench, for the following reasons:**
1) there can be performance gaps between the dev set and test set, as seen in other models on the current leaderboard (e.g., Distillery + GPT-4o has 67.21 on the dev set but 71.83 on the test set, while Insights AI has 72.16 on the dev set but 70.26 on the test set), and the official leaderboard uses test set metric as the main ranking metric.
2) the current reported number only indicates marginal improvement on the dev set. It's hard to justify the superiority of the performance with statistical significance.

I understand the concern mentioned by the authors in the paper that
> due to the anonymity policy, we only report the execution accuracy on the development dataset

However, according to the official Author Guide of ICLR 2025, it is permitted to report such numbers: https://iclr.cc/Conferences/2025/AuthorGuide
> Q: Can you explain how to treat de-anonymization in the case where a submitted paper refers to a challenge they won which can identify the authors?
> It is ok to report the results on the leaderboard of a challenge. The authors can include the ranking and the name of the challenge. The reviewers will be advised to not intentionally search the authors by examining the leaderboard.

**In Table 3, I recommend revising the ablation configurations or explaining clearly the rationales behind the choice**

I'm confused by the current configurations. By comparing L435-436 I understand that UQU is providing a positive performance gain, but what does L436 vs 437 mean? What is the generation backbone model if it is not GPT-4o? I would recommend keeping the backbone model consistent (e.g., always using GPT-4o) while ablating the modules.

In addition, I think the configuration "revision + generation" might be worth adding since that is also a common approach. I do not think entity retrieval is always done in NL2SQL approaches.

**In writing, I recommend removing the statements that the proposed approach outperformed SOTA models on both datasets**

I think it is fair to claim that the proposed method outperformed comparable (criterion: open source or published paper) models, but it would be misleading to say it outperformed SOTA models, especially on the Spider dataset.

### **1.2 For 3, I'm mainly referring to Table 2 in the paper.**

The main issue is that the evaluation process of getting these metrics is not included in the paper. Excluding the context makes the numbers hard to interpret. What does accuracy = 0.8 mean? Likewise for Time (s) and Cost.



## **2. The presentation can be clearer**

**Table headers can map to sections**

The modules in Figure 4 align with Section 4 well. However, the granularity in Table 3 seems not uniform. Are "Revision" and "Generation(*)" both part of section 4.3? Consider splitting them into columns or color-coding them for better readability.

Same thing with Table 4, which is not very straightforward how components in each row map to the framework anatomy and which rows to compare.

**The paper has some inconsistencies/unclarities that should be addressed.**

One thing that confused me at the beginning was "LAIA-NLUer", which seems to only appear at the beginning and end of the paper. My assumption is it is later named the "UQU" module. I would recommend unifying that.

L407 mentioned Qwen-1.5-Coder, yet I don't think it is included.


**Typos and formatting**

L081-L083, among others: CHESS Talaei et al. (2024) -> CHESS (Talaei et al., 2024)

L236: 'unsatisfactory' -> ``unsatisfactory''

L301: "SELECT" -> ``SELECT''

L375: Bird -> BIRD

L435: Generaton -> Generation

**Questions:**

Please refer to the Weakness section.

---

> ### Author Response · Authors · 2024-11-18
>
> We thank the reviewer for their time and interest in our work. The detailed instructions provided helpful feedback to improve our work. We have addressed the reviewer’s questions and suggestions, and we provide a list of responses below:
>
> ---
>
> **Q1.1.1 BIRD Benchmark Reporting:**
>
> **A1.1.1** We have submitted our code to the BIRD dataset team and are currently awaiting their results. We understand the importance of reporting the official test scores and will update our paper accordingly once we receive the results. This may take some time, and we appreciate your patience.
>
> ---
>
> **Q1.1.2 Ablation Study in Table 3:**
>
> **A1.1.2** We have revised Table 3 to ensure consistency by using GPT-4 as the backbone model across all configurations. The updated results are as follows:
>
>    | Method | Dev EX |
>    |-----------|-----------|
>    | UQU + Entity Retrieval + SQL Generation (GPT-4) + Revision (GPT-4)  | 60.36 |
>    | Entity Retrieval  + SQL Generation (GPT-4) + Revision (GPT-4) | 55.28 |
>    | Entity Retrieval + SQL Generation (GPT-4)  | 51.25 |
>    | SQL Generation (GPT-4)  | 46.35 |
>
>    We hope this clarifies the rationale behind our choices and addresses the confusion.
>
> ---
>
> **Q1.1.3 Configuration "Revision + Generation":**
>
> **A1.1.3** We understand that entity retrieval is not always utilized in NL2SQL approaches, and we appreciate you pointing this out. Moving forward, we will place greater emphasis on user question understanding, SQL generation, and revision in our research. Your feedback is invaluable in helping us improve our work.
>
> ---
>
> **Q1.1.4 Statements on Performance:**
>
> **A1.1.4** We have revised our statements to accurately reflect the performance of our method in the paper. The updated phrasing is: "Overall, our framework performs better compared to currently available open-source or published methods."
>
> ---
>
> **Q1.2** For 3, I'm mainly referring to Table 2 in the paper.
> The main issue is that the evaluation process of getting these metrics is not included in the paper. Excluding the context makes the numbers hard to interpret. What does accuracy = 0.8 mean? Likewise for Time (s) and Cost.
>
> **A1.2** We have revised our paper accordingly to provide clearer context for the metrics presented in Table 2. Specifically, we have updated the term 'accuracy' to 'dev EX' to maintain consistency with Table 1. In addition, we have included a detailed explanation in Section 5.1.
>
> Here's a clarification of the metrics:
> - **dev EX** represents the accuracy of generating the final SQL for 10 questions.
> - **Time (s)** indicates the total time required to generate the final SQL for these 10 questions.
> - **Cost** corresponds to the monetary cost incurred to generate the final SQL for the 10 questions.

---

> > ### Author Response · Authors · 2024-11-18
> >
> > **Q2.1 Table headers can map to sections**
> >
> > **A2.1** We have updated Table 3 to change "Generation" to "SQL Generation" to maintain consistency with Figure 4 and section 4.3.
> >
> > In response to your feedback on Table 4, we have broken down the original two columns into five for better granularity and readability. The updated table is as follows:
> >
> > | UQU | Entity Retrieval | Generation | Dev EX |
> > |-----------|-----------|-----------|-----------|
> > | GPT-4o-mini (finetuned) | MinHASH + Stella-400M  | GPT-4o | 67.28 |
> > | Mistral-7B (finetuned) | MinHASH + Stella-400M  | GPT-4o | 65.16 |
> > | GPT-4 | MinHASH + Stella-400M | GPT-4o | 59.62 |
> > | GPT-4 | MinHASH + Stella-400M | DeepSeek-Coder-V2-Instruct (finetuned) | 55.78 |
> > | GPT-4 | MinHASH + Stella-400M | DeepSeek-Coder-V2-Base (finetuned) | 50.41 |
> > | GPT-4 | MinHASH + Stella-400M | Qwen-1.5-Coder (finetuned) | 30.82 |
> > | GPT-4 | MinHASH + Stella-400M | GPT-4 | 53.17 |
> > | GPT-4 | MinHASH + Stella-1.5B | GPT-4 | 51.36 |
> > | GPT-4 | MinHASH + text-embedding-3-large | GPT-4 | 51.25 |
> > | GPT-4 | BM25 + text-embedding-3-large | GPT-4 | 49.34 |
> >
> > ---
> >
> > **Q2.2 The paper has some inconsistencies/unclarities that should be addressed.**
> >
> > **A2.2** Concerning the ambiguity between "LAIA-NLUer" and "UQU," we have clarified that UQU is the name of the module, while LAIA-NLUer is a model contained within the UQU module. We believe they should not be merged for this reason. Additionally, we previously omitted the results for Qwen-1.5-Coder due to space constraints, but we have now included these results in Table 4.
> >
> > ---
> >
> > **Q2.3 Typos and formatting**
> >
> > **A2.3** The formatting and citation issues have been corrected. Specifically:
> >    - CHESS Talaei et al. (2024) has been changed to CHESS (Talaei et al., 2024).
> >    - The quotation marks around 'unsatisfactory' and "SELECT" have been removed.
> >    - We have corrected the typo "Generaton" in Table 4 to "Generation."

---

> > ### Comment · Reviewer_9hYg · 2024-11-23
> >
> > Thank you for your response. It would be great to send a further update here once you hear back from the BIRD leaderboard or give an estimate on when it will be available.

---

### Official Review · Reviewer_7reZ · 2024-11-04

**Soundness:** 2
**Presentation:** 2
**Contribution:** 2
**Rating:** 3
**Confidence:** 4

**Summary:**

This paper addresses the problem of question answering from multiple tables by converting users’ natural language questions into executable SQL, a task at the intersection of TableQA and text-to-SQL. The paper focuses on retrieval-augmented text-to-SQL methods, where parts of the tables are retrieved to better form the final SQL query. It proposes to tackle this task in several steps:

1. Understanding the User Query (NLU): Extract keywords and break down the query into simpler steps.
2. Table Retrieval: Identify and extract the necessary data from the underlying tables.
3. SQL Writing Stage: Create the SQL query based on the retrieval output, and optionally revise it using feedback from SQL syntax error messages.

The authors propose a dataset for query decomposition and keyword extraction in the multi-table text-to-SQL task, called _LAIA-NLU_. This dataset is constructed using questions from the BIRD dataset, and human-verified GPT-4o labels. Furthermore, they introduce _LAIA-SQL_, a retrieval-augmented text-to-SQL system that implements these three stages.

The paper experiments with various configurations of LAIA-SQL, using:

- NLU Stage: Few-shot `GPT-4`, fine-tuned `GPT-4o-mini`, and fine-tuned `Mistral-7B`.
- Table Retrieval: BM25 and MinHASH combined with OpenAI’s `text-embedding-3`, `Stella-400M`.
- SQL Writing Stage: Few-shot `GPT-4`, `GPT-4o`, and fine-tuned `DeepSeek-Coder` models.

Experimental results on the BIRD and Spider datasets show that LAIA-SQL outperforms prior systems in terms of lower latency and cost, and higher execution accuracy.

**Strengths:**

Evaluating latency and cost of text-to-SQL systems is of value but is often ignored in research. It is great that this paper is paying attention to this.

**Weaknesses:**

## 1. The connection between this paper and prior work should be improved.
The paper revisits concepts that have been extensively studied in the field. Question decomposition has been thoroughly explored in TableQA, knowledge base QA [1], and more general agentic search systems [4][5]. Additionally, similar research [2] [3] investigates prompting LLMs for query decomposition in text-to-SQL. Table retrieval for text-to-SQL is studied in [7]. These references are not included in the problem definition, prior work section, and experiments, and it would be beneficial to address this.

Other areas where the connection could be improved include:
- The paper assumes that query decomposition and keyword extraction are the primary methods for solving multi-table QA (L048, among others), which is inaccurate.
- Section 2.2 seems to conflate the NLU and TableQA literatures.
- L078 states, "There is a lack of quantitative evaluation metrics for assessing NLU performance across different LLMs within the TableQA domain," which is not substantiated. More broadly, what is the need for a question decomposition dataset, beyond the likes of [1]?
- The reference to TA-SQL (Line 082) should be corrected to [6].
- L121 mentions GraphRag, but that does not experiment with tables.

I recommend conducting a more comprehensive literature review, including papers published prior to 2024, and revising the claims and experiments accordingly.

## 2. The comparison of experimental results with prior work is potentially misleading.

Importantly, the comparison of experimental results with prior work is not apples-to-apples. For instance, the comparison against CHESS uses the evaluation result from that paper. However, CHESS uses `GPT-4-Turbo`, which is slower and more expensive than `GPT-4o` used in this paper. This undermines the claim that this paper outperforms prior work on cost and latency. Additionally, latency and cost (reported in Table 2) are not measured in the same end-to-end setting as Table 1.

## 3. Experiments could benefit from additional clarity and details.
The paper presents several ablation results, but the purpose of the ablation studies and the conclusions that can be drawn from them are unclearer. For example, it would be helpful to understand whether the reported accuracy and latency/cost benefits are due to better decomposition of the task compared to agentic approaches, better/shorter prompts, or higher quality of the models that are used and fine-tuned.

Some details that need further elaboration on include:
- In Section 4.2, it would be helpful to explain how tables are encoded into vectors and what preprocessing is done, considering that tables can be quite large.
- In the results reported in Tables 1 and 2, it should be specified which embedding model and reranker are used.
- In Tables 3 and 4, why do systems alternate between GPT-4 and GPT-4o? Keeping all but one of the components the same would help make better sense of the ablation results.

**References:**

1. BREAK It Down: A Question Understanding Benchmark

2. Exploring Chain of Thought Style Prompting for Text-to-SQL

3. Semantic Decomposition of Question and SQL for Text-to-SQL Parsing

4. DARA: Decomposition-Alignment-Reasoning Autonomous Language Agent for Question Answering over Knowledge Graphs

5. Iterated Decomposition: Improving Science Q&A by Supervising Reasoning Processes

6. Before Generation, Align it! A Novel and Effective Strategy for Mitigating Hallucinations in Text-to-SQL Generation

7. Towards Robustness of Text-to-SQL Models Against Natural and Realistic Adversarial Table Perturbation

**Questions:**

1. What is the "Accuracy" metric in Table 2? How does it differ from dev EX in Table 1?
1. In Figure 1, a comparison is shown between GPT-4o and the "ours" method for task decomposition and keyword extraction. However, from what I understand, "ours" also uses GPT-4o. What is the main difference?

---

> ### Author Response · Authors · 2024-11-17
>
> We thank the reviewer for their time and interest in our work. The detailed instructions provided helpful feedback to improve our work. We have addressed the reviewer’s questions and suggestions, and we provide a list of responses below:
>
> ---
>
> **Q1 The connection between this paper and prior work should be improved.**
>
> **A1** We have add the previous work in related work, here is the content in the revised manusript:
>
> "In the domain of table QA, researchers have also developed several methods for NLU, such as QDecomp Tai et al. (2023) and QPL Eyal et al. (2023), which investigate prompting LLMs for query decomposition. Meanwhile, methods like DARA Fang et al. (2024), Iterated Decomposition Reppert et al. (2023) show potentials for NLU. Additionally, datasets have been constructed, such as BREAK Wolfson et al. (2020). However, existing NLU datasets in the table QA domain tend to decompose tasks into very fine-grained components, thereby overlooking the primary requirements of the task. This approach lacks a comprehensive evaluation of the model’s capabilities in task decomposition. Moreover, applications of table QA require assessment of the model’s ability to extract relevant keywords from user queries related to database filter values, column names, or table names, a capability often missing from previous datasets."
>
> **A1.1**Clarification on Query Decomposition and Keyword Extraction:
>
> Our original statement was "Effective SQL generation requires the model to excel in natural language understanding (NLU), which can be divided into two areas: 1) fine-grained task decomposition and 2) precise keyword extraction."
> We did not intend to imply that keyword extraction and task decomposition are the primary methods for solving multi-table QA. Our main point is that improving SQL generation accuracy necessitates the model's strong performance in natural language understanding.
>
> **A1.2** Regarding Section 2.2:
>
> We appreciate your feedback. Our intention was to provide a comprehensive overview of the developments in both general NLU and TableQA fields. We recognize that a good related work section should summarize the progression in the broader domain as well as in the specific sub-field. We aimed to highlight the advancements and existing challenges in NLU both generally and within the context of table QA.
>
> **A1.3** Quantitative Evaluation Metrics for NLU:
>
> We acknowledge that our initial phrasing may not have been precise. Upon careful review, we have revised our statement to better reflect our analysis of existing research:
>    "Although the BREAK dataset \cite{wolfson2020break} includes QA pairs that decompose questions in a highly granular manner, it lacks data pertinent to the primary task. Additionally, while it extracts keywords, these are neither comprehensive nor noise-free, making them unsuitable for evaluating the NLU performance in table QA."
>
> **A1.4** Reference Correction for TA-SQL:
>
> We have corrected the reference to TA-SQL to [6] as suggested.
>
> **A1.5** Clarification on GraphRAG:
>
> Our intention was to indicate that some researchers have applied the GraphRAG method to TableQA with notable success. To avoid any confusion, we have decided to remove this part from our text.

---

> ### Author Response · Authors · 2024-11-17
>
> **Q2. The comparison of experimental results with prior work is potentially misleading.**
>
> **A2** As mentioned in Table 2's caption, we used GPT-4o as the base model to test methods including CHESS and TA-SQL. Specifically, we replaced CHESS’s default GPT-4-Turbo with GPT-4o for a fair comparison. Under this setting, CHESS incurred a cost of 11 USD for 10 questions, whereas our method required only 0.32 USD. This demonstrates that, with the same base model and dataset, our approach outperforms others in terms of time, accuracy, and cost.
>
> The results in Table 1 are sourced from BIRD and the Spider leaderboard. Testing all methods using GPT-4o would indeed be cost-prohibitive. Therefore, in Table 2, we tested different methods on the same 10 questions using the same base model (GPT-4o), showing that our method achieved the best accuracy.
>
> Additionally, some methods mentioned in Table 1, such as Dail-SQL, were not included in Table 2 due to unsuccessful reproduction. For instance, the reproduction of Dail-SQL took 756 seconds for 10 questions with an accuracy of just 0.3, potentially leading to ambiguous results.

---

> > ### Author Response · Authors · 2024-11-17
> >
> > **Q3. Experiments could benefit from additional clarity and details.**
> >
> > **A3**
> >
> > **1.Purpose of Ablation Studies:**
> >
> > We appreciate your suggestion to explore the contributions of different task decomposition methods, prompt strategies, and fine-tuned models on accuracy, latency, and cost. Indeed, this is a fascinating area for future research. In our current ablation studies, we aimed to control for variables by using fine-tuned models and API calls to provide consistent comparisons. We uniformly utilized GPT-4o as the base model in Table 2 to ensure fair comparisons and reduce the impact of external factors such as GPU capabilities and network speed.
> >
> > **2.Encoding of Tables:**
> > In Section 4.2, we have added a detailed explanation of how tables are encoded into vectors and the preprocessing steps involved. This includes handling the large size of tables efficiently. We have now included detailed information in the revised manuscript:
> >
> > "MinHash creates embeddings by generating a fixed-size signature (a vector) for a set, capturing its similarity with other sets through hash functions that approximate the Jaccard similarity between them. BM25 does not create embeddings, instead, it is a probabilistic retrieval model using term frequency and inverse document frequency to score the relevance of documents to a query."
> >
> > **3.Embedding Models and Rerankers:**
> > Due to space constraints, we did not specify the models used in Tables 1 and 2 in the original submission. We have now included detailed information in the revised manuscript:
> >
> > | Method | Model | Dev EX | Time | Cost |
> > | --- | --- | --- | --- | --- |
> > | Distillery | Fine-tuned GPT-4o | 67.21 | - | - |
> > | CHESS | Minhash+Fine-tuned DeepSeek+GPT-4-turbo | 65.00 | 87.2 | - |
> > | SFT CodeS-15B | - | 58.47 | 84.9 | 79.4 |
> > | Dail-SQL | GPT-4 | 54.76 | 84.4 | 86.6 |
> > | MAC-SQL | Fine-tuned GPT-4 + Oracle Schema | 57.56 | 86.7 | 82.8 |
> >
> > **4.Ablation Study on Models:**
> > In Table 3, we compared different modules with an ablation study, including results for GPT-4. Table 4 focuses on the impact of using different models. We have revised these tables to clarify the components and their effects on accuracy.

---

> ### Author Response · Authors · 2024-11-17
>
> **Q4** What is the "Accuracy" metric in Table 2? How does it differ from dev EX in Table 1?
>
> **A4** We have updated the "Accuracy" metric in Table 2 to "dev EX" for consistency with Table 1.
>
> ---
>
> **Q5** In Figure 1, a comparison is shown between GPT-4o and the "ours" method for task decomposition and keyword extraction. However, from what I understand, "ours" also uses GPT-4o. What is the main difference?
>
> **A5** In Figure 1, GPT-4o refers to the base model. "Ours" represents the LAIA-NLUer model, which has been fine-tuned using the LAIA-NLU dataset for task decomposition and keyword extraction. We have revised the figure to clarify this distinction.
>
> ---
>
> Thank you again for your suggestions. We understand your concerns about the motivation and complexity of our framework. It is important to emphasize that our main contribution lies in the **Dataset Track** rather than direct competition with methods like CHESS. Our dataset, LAIA-NLU, has undergone three rounds of meticulous human verification, ensuring its high quality. This required significant human and material resources. This dataset addresses the existing gap in the field of table QA by providing validation data specifically for fine-grained multi-level task decomposition and keyword extraction.
>
> Experimental results demonstrate that LAIA-NLU can effectively evaluate different models' understanding of user questions. Additionally, models fine-tuned on our dataset exhibit substantial improvements in SQL generation accuracy, resulting an improvement of accuracy from 59.62% to 67.28%. We believe LAIA-NLU has broad applicability, potentially benefiting future NL2SQL algorithm development and other applications involving user question understanding.
>
> Thank you again for your valuable feedback. We believe that a positive and collaborative discussion can greatly contribute to the advancement of our field. We look forward to engaging in a meaningful dialogue to further improve our research! Have a nice day.

---

> ### Author Response · Authors · 2024-11-25
>
> Dear Reviewer 7reZ:
>
> Thanks again for all of your constructive suggestions, which have helped us improve the quality and clarity of the paper!
>
> Since the author-reviewer discussion period will end soon on Nov 26th, we appreciate it if you take the time to read our comments and give us some feedback. Please don't hesitate to let us know if there are any additional clarifications or experiments that we can offer, as we would love to convince you of the merits of the paper. We appreciate your suggestions. If our response resolves your concerns, we kindly ask you to consider raising the rating of our work.
>
> Thanks for your time and efforts!
>
> Best,
>
> Authors of Paper 10678

---

> ### Author Response · Authors · 2024-11-26
> **Sincerely Looking Forward to More Discussions**
>
> Dear Reviewer 7reZ:
>
> Thank you for your valuable feedback. We have carefully addressed all your concerns and deeply appreciate your thoughtful suggestion to revise the manuscript. As a submission in the dataset track, our paper introduces a dataset that has undergone multiple rounds of manual verification to ensure high quality, effectively addressing the limitations of existing datasets of a similar nature. Additionally, fine-tuning models with our dataset significantly enhances their natural language understanding capabilities. This improvement, in turn, boosts the accuracy of SQL query generation by the NL2SQL method, as well as increases efficiency and reduces associated costs. In response to the valuable feedback received, we have also incorporated additional experiments as suggested, thereby making our study more comprehensive and robust. We hope these revisions comprehensively address your comments and are happy to discuss any additional concerns you may have.
>
> Best,
>
> Authors of Paper 10678

---

### Official Review · Reviewer_oXTV · 2024-11-05

**Soundness:** 3
**Presentation:** 4
**Contribution:** 3
**Rating:** 8
**Confidence:** 3

**Summary:**

The paper addresses challenges in transforming natural language questions into SQL commands for multi-table question answering (Table QA). The authors introduce LAIA-NLU, a dataset focused on understanding natural language in NL2SQL tasks through task decomposition and keyword extraction. This dataset contains 1,500 carefully curated QA pairs, enabling improved interpretation of user intent in table-based queries. Building on this, the authors developed LAIA-NLUer, a model that enhances user question understanding, and LAIA-SQL, a retrieval-augmented NL2SQL framework optimized for cost, speed, and accuracy.

**Strengths:**

- Paper is well written and easy to follow.
- Their method is novel, efficient and effective.

**Weaknesses:**

- Would like to see some Human Evaluation done on the outputs.
- The authors mention this in the limitations as well but maybe the dataset could be expanded using data augmentation strategies.

**Questions:**

- I am curious to see some examples of the revision module correcting mistakes.

---

> ### Author Response · Authors · 2024-11-18
>
> We thank the reviewer for their time and interest in our work. The detailed instructions provided helpful feedback to improve our work. We have addressed the reviewer’s questions and suggestions, and we provide a list of responses below:
>
> ---
>
> After carefully analyzing all the revisions, we observed that there were significant improvements in the accuracy of examples. Specifically:
>
> - 17.74% of the successful cases resolved the "no such column" issue, which involved correcting references to tables and columns in the SQL code.
>
> - 4.84% of the successful cases addressed the "Near 's': Syntax Error" issue by changing the filtering value to the correct format.
>
> - 1.61% of the successful cases fixed the "No Such Function" problem, where there was an incorrect function call in the SQL (e.g., using the `strftime` function instead of the `year` function, which does not exist).
>
> - 75.81% of the successful cases resulted in correct SQL sentences after being generated a second time.
>
> Here are specific examples illustrating these improvements:
>
> **### Example One**
>
> **SQL Generation**
>
> - Result:`"no such column: E1.link_to_event"`
>
> - Question: "Based on the total cost for all event, what is the percentage of cost for Yearly Kickoff event?"
>
> - Evidence: "percentage = DIVIDE(SUM(cost where event_name = 'Yearly Kickoff'), SUM(cost)) * 100"
>
> - GOLD_SQL: "SELECT CAST(SUM(CASE WHEN T1.event_name = 'Yearly Kickoff' THEN T3.cost ELSE 0 END) AS REAL) * 100 / SUM(T3.cost) FROM event AS T1 INNER JOIN budget AS T2 ON T1.event_id = T2.link_to_event INNER JOIN expense AS T3 ON T2.budget_id = T3.link_to_budget"
>
> - PREDICTED_SQL: "SELECT (SUM(E1.cost) * 100.0 / (SELECT SUM(E2.cost) FROM expense E2)) AS percentage FROM expense E1 JOIN event EV ON `E1.link_to_event` = EV.event_id WHERE EV.event_name = 'Yearly Kickoff';"
>
> **Revision**
>
> - Result:"correct answer"
>
> - Revised_SQL: "SELECT (SUM(E1.cost) * 100.0 / (SELECT SUM(E2.cost) FROM expense E2)) AS percentage FROM expense E1 JOIN budget B ON E1.link_to_budget = B.budget_id JOIN event EV ON `B.link_to_event` = EV.event_id WHERE EV.event_name = 'Yearly Kickoff';"
>
> ---
>
> **### Example Two**
>
> **SQL Generation**
>
> - Result:`"near\"s\": syntax error"`
>
> - Question: "Among the students from the Student_Club who attended the event\"Women's Soccer\", how many of them want a T-shirt that's in medium size?"
>
> - Evidence: "Women's Soccer is an event name; T-shirt that is in medium size refers to t_shirt_size = 'Medium'"
>
> - GOLD_SQL: "SELECT COUNT(T1.event_id) FROM event AS T1 INNER JOIN attendance AS T2 ON T1.event_id = T2.link_to_event INNER JOIN member AS T3 ON T2.link_to_member = T3.member_id WHERE T1.event_name = 'Women''s Soccer' AND T3.t_shirt_size = 'Medium'"
>
> - PREDICTED_SQL: "SELECT COUNT(*) AS medium_tshirt_count FROM attendance T1 JOIN event T2 ON T1.link_to_event = T2.event_id JOIN member T3 ON T1.link_to_member = T3.member_id WHERE T2.event_name = `'Women's Soccer'` AND T3.t_shirt_size = 'Medium';"
>
> **Revision**
>
> - Result:"correct answer"
>
> - Revised_SQL: "SELECT COUNT(*) AS medium_tshirt_count FROM attendance T1 JOIN event T2 ON T1.link_to_event = T2.event_id JOIN member T3 ON T1.link_to_member = T3.member_id WHERE T2.event_name = `'Women''s Soccer'` AND T3.t_shirt_size = 'Medium';"
>
> ---
>
>
> **### Example Three**
>
> **SQL Generation**
>
> - Result:`"no such function: year"`
>
> - Question: "List all patients who first came to the hospital after year 1992 with prothrombin time (PT) level that are normal."
>
> - Evidence: "first came to the hospital after year 1992 refers to year(First Date) > 1992; prothrombin time (PT) level that are normal refers to PT < 14;"
>
> - GOLD_SQL: "SELECT T1.ID FROM Patient AS T1 INNER JOIN Laboratory AS T2 ON T1.ID = T2.ID WHERE STRFTIME('%Y', T1.First Date) > '1992' AND T2.PT < 14"
>
> - PREDICTED_SQL: "SELECT T1.ID FROM Patient T1 JOIN Laboratory T2 ON T1.ID = T2.ID WHERE `year(T1.First Date)` > 1992 AND T2.PT < 14;"
>
> **Revision**
>
> - Result:"correct answer"
>
> - Revised_SQL: "SELECT T1.ID FROM Patient T1 JOIN Laboratory T2 ON T1.ID = T2.ID WHERE `strftime('%Y', T1.First Date)` > '1992' AND T2.PT < 14;"
>
> ---
>
> **### Example Four**
>
> **SQL Generation**
>
> - Result: "incorrect answer"
>
> - Question: List circuits which host 4 f1 races from year 1990 to 2000."
>
> - Evidence: "first came to the hospital after year 1992 refers to year(First Date) > 1992; prothrombin time (PT) level that are normal refers to PT < 14;"
>
> - GOLD_SQL: "SELECT T1.ID FROM Patient AS T1 INNER JOIN Laboratory AS T2 ON T1.ID = T2.ID WHERE STRFTIME('%Y', T1.First Date) > '1992' AND T2.PT < 14"
>
> - PREDICTED_SQL: "SELECT T1.ID FROM Patient T1 JOIN Laboratory T2 ON T1.ID = T2.ID WHERE year(T1.First Date) > 1992 AND T2.PT < 14;"
>
> **Revision**
>
> - Result: "correct answer"
>
> - Revised_SQL: "SELECT T1.ID FROM Patient T1 JOIN Laboratory T2 ON T1.ID = T2.ID WHERE strftime('%Y', T1.First Date) > '1992' AND T2.PT < 14;"

---

> > ### Comment · Reviewer_oXTV · 2024-11-27
> > **Response to Review**
> >
> > Thank you for your response. I am inclined to keep my score as I already significantly liked your work.

---

### Note · Authors · 2024-12-13

I have read and agree with the venue's withdrawal policy on behalf of myself and my co-authors.